# Wildfire Detection via a Dual-Channel CNN with Multi-Level Feature Fusion

Zhiwei Zhang [1], Yingqing Guo [1,*], Gang Chen [1] and Zhaodong Xu [2]

1   College of Mechanical and Electronic Engineering, Nanjing Forestry University, Nanjing 210037, China; zhiwei_zhang@njfu.edu.cn (Z.Z.); chengang8059@njfu.edu.cn (G.C.)
2   China-Pakistan Belt and Road Joint Laboratory on Smart Disaster Prevention of Major Infrastructures, Southeast University, Nanjing 210096, China; zhdxu@163.com
*   Correspondence: gyingqing@126.com; Tel.: +86-13951835602

**Abstract:** Forest fires have devastating impacts on ecology, the economy, and human life. Therefore, the timely detection and extinguishing of fires are crucial to minimizing the losses caused by these disasters. A novel dual-channel CNN for forest fires is proposed in this paper based on multiple feature enhancement techniques. First, the features' semantic information and richness are enhanced by repeatedly fusing deep and shallow features extracted from the basic network model and integrating the results of multiple types of pooling layers. Second, an attention mechanism, the convolutional block attention module, is used to focus on the key details of the fused features, making the network more efficient. Finally, two improved single-channel networks are merged to obtain a better-performing dual-channel network. In addition, transfer learning is used to address overfitting and reduce time costs. The experimental results show that the accuracy of the proposed model for fire recognition is 98.90%, with a better performance. The findings from this study can be applied to the early detection of forest fires, assisting forest ecosystem managers in developing timely and scientifically informed defense strategies to minimize the damage caused by fires.

**Keywords:** wildfire hazard; wildfire detection; dual-channel CNN; multi-level feature fusion

## 1. Introduction

Forests are often referred to as the "lungs of the Earth" due to their two important values. One is their visible economic value, and the other is their intangible ability to regulate the climate and maintain ecological balance. Forest fires have occurred frequently in recent years due to extreme events such as lightning, volcanic eruptions, and human activities [1–4]. For example, in 2015, forest fires in Indonesia burned over 2.6 million hectares of land, resulting in more than USD 16 billion in economic losses, and posing significant threats to the local biodiversity and endangered species [5,6]. Since September 2019, the prolonged Australian bushfires have burned over 10 million hectares of land, causing at least 33 deaths and displacing billions of animals [7,8]. Based on the analysis conducted by Lukić et al. [9] in 2012, the frequency of fires in the Tara Mountains exceeded the average level of fire occurrence in Serbia, primarily due to the strong influence of climatic conditions [10]. The occurrence of forest fires and the extent of the damage they inflict are also influenced by factors connected to the forest itself [11–13], such as the species of trees present [14], the combustibility of the forest fuels, and the water content [15,16]. Therefore, adopting proactive and effective forest fire monitoring methods is crucial for ecological, social, and economic reasons [17–19]. The use of image processing and deep learning methods has become the mainstream research direction in this field [20].

Traditional image-based fire detection methods employ feature extraction algorithms based on prior knowledge to analyze flame or smoke characteristics, and machine learning algorithms are then used to determine if a fire has occurred. Among these methods, flame or smoke features are analyzed under various color spaces [21] such as RGB [22], YCbCr [23],

and YUV [24], or multiple features related to flame or smoke are fused (e.g., motion information [25], shape [26], and area [27]) to construct expert systems for fire detection. In 2009, Rudz et al. [28] accurately identified fire characteristics and achieved better fire detection through the use of fire color space and segmented fire images. In 2019, Matlani et al. [29] improved the accuracy of detection by integrating features such as Haar, SIFT, and others. Some researchers construct fire models by evolving dynamic textures with multiple spatiotemporal features [30] or by detecting color changes in fire motion regions [31] and irregular boundaries [32], among others. In 2013, Wang et al. [33] developed prior flame probability by combining flame color probability and the results of the Wald–Wolfowitz randomness test. This method was shown to have good robustness and the ability to adapt to different environments, making it a reliable and practical approach to flame detection.

Forest fires can be roughly divided into the following five stages: the ignition stage, the propagation stage, the peak burning stage, the stage of fire suppression and weakening, and the stage of fire termination [34]. Each stage has distinct fire characteristics [35]. For example, during the ignition stage [36], the fire may only produce faint smoke, while in the propagation stage, the fire can spread rapidly. At the peak burning stage, flames and smoke may become very thick, and the fire can reach its maximum intensity. Therefore, fire monitoring algorithms that are based solely on empirical knowledge may only perform well in specific scenarios and may have limited generalization to other situations [37].

In contrast, convolutional neural networks (CNNs) automatically extract features from provided data [38,39], thus avoiding the limitations of manually selected features [40,41]. Furthermore, an excellent CNN can also achieve good results in other application domains. In 2019, Kim et al. [42] and Lee et al. [43] both utilized Faster-RNN to identify fire regions. The former directly recognized the fire and smoke characteristics, while the latter used a combination of local and global frame features as the judgment basis. In 2017, Wang et al. [44] replaced the CNN's fully connected layer with SVM to obtain better detection results. In 2016, Frizzi et al. [45] significantly reduced the time cost by designing a feature map instead of the original frame. In 2020, Liu et al. [46] proposed a fire recognition model based on a two-level classifier. They used a first-level classifier composed of HOG and Adaboost for preliminary recognition and a second-level classifier composed of CNN and SVM for further fire recognition with a higher accuracy. In 2022, Guo et al. [47] designed a Lasso_SVM layer to replace the fully connected layer in the original model and improved the model detection accuracy through segmented training. In the same year, Qian et al. [48] proposed a weighted fusion algorithm for forest fire source recognition based on two weakly supervised models, Yolov5 and EfficientDet. In 2020, Xie et al. [49] designed a network called Controllable Smoke Image Generation Neural Networks-V2 (CSGNet-v2) to generate realistic smoke images based on smoke characteristics, which can be used for smoke detection in forest fires. In 2021, Ding et al. [50] proposed a dual-stream convolutional neural network based on attention mechanisms, which pays more attention to the spatiotemporal characteristics of smoke and enhances the ability to segment and recognize small smoke particles.

In addition to simply improving the CNN structure, some researchers improved their algorithms by complementing the advantages of traditional image processing methods with those of CNNs. For example, in 2022, Yang et al. [51] optimized the method for identifying early spring green tea by using semi-supervised learning and image processing. Wu et al. [52] proposed an adaptive deep-learning flame and smoke classifier based on traditional feature extraction algorithms in the Caffe framework in 2017. In 2019, Wang et al. [53] combined the advantages of traditional image processing and CNNs to propose an adaptive pooling convolutional neural network that effectively extracts features by pre-learning the flame segmentation area features and avoiding the blind nature of traditional feature extraction, thus improving the effectiveness of CNN learning. In 2023, Zheng et al. [54] effectively extracted target features by using cross-attention blocks to capture differences in global information and local color and texture feature information.

The accuracy and generalizability of CNN-based methods are superior to the suggested fire detection algorithms that are based on a priori knowledge. A novel two-channel CNN model is put forth in this article, and the basic network is optimized using two feature fusion techniques and the addition of an attention mechanism. A more potent dual-channel network is created by fusing two single-channel networks that have varying input sizes but similar structural characteristics. Migration learning provides a solution to the issue of the model training's simple overfitting and lowers the model training's time cost. The research methodology outlined in this paper can be applied to the task of early detection and identification of forest fires. Its findings will assist forest ecosystem managers in gaining timely awareness of forest fires and implementing scientific fire prevention measures to mitigate the losses caused by such fires. This is crucial for the protection of forest resources.

## 2. Materials and Methods

### 2.1. Construction of the Experimental Dataset

The dataset plays a crucial role in deep learning research and is one of the key factors in achieving exceptional model performance. In this study, a dataset of 14,000 images was utilized, which was obtained by members of the research team through the Google search engine. The images can be categorized into the following two groups: "non-fire" and "fire" [47]. The "fire" category encompasses images of flames, white smoke, black smoke, dense smoke, and thin smoke generated by fires in different settings, such as forests, grasslands, fields, and urban areas. The dataset creation method was inspired by references [17,55,56], which involved including various angles of regular forest images in the "non-fire" category, as well as incorporating more complex and disruptive images, such as clouds, sunlight, and pyrocumulus clouds, under different weather conditions. By introducing these disruptive factors, the objective was to enhance the model's ability to recognize smoke and flames and improve its robustness. Following an approximate 7:3 ratio, the images from the different categories in the dataset were divided into training and testing sets, with 10,000 images in the training set and the remaining images used for testing. Figure 1 displays some sample images that can facilitate a better understanding of the dataset's composition and characteristics.

### 2.2. Essential Basic Knowledge

Before building the network model in this article, it is necessary to introduce some fundamental concepts, including feature fusion, transfer learning, and attention mechanism.

#### 2.2.1. Feature Fusion

Feature fusion is a widely used technique applied in fields such as computer vision and natural language processing. It involves combining feature information from diverse sources through concatenation, merging, stacking, and cascading operations. This technique effectively tackles issues such as data sparsity and noise interference, and enhances feature expression and generalization ability [57–61].

This article presents two feature fusion methods. The first method is inspired by the SPP idea, which replaces a single pooling layer in the base network with a combination of a max pooling layer and an average pooling layer that acts simultaneously and fuses the results of the two pools. The second approach fuses the shallow and deep features of the network and is influenced by the idea of residual networks. These methods enhance the richness of feature information, enabling the model to learn data features from multiple perspectives and provide a more comprehensive and accurate description of the data's essence. Based on the inspiration from SPP and residual structures, two network architectures for feature fusion methods were designed in this study, as shown in Figure 2.

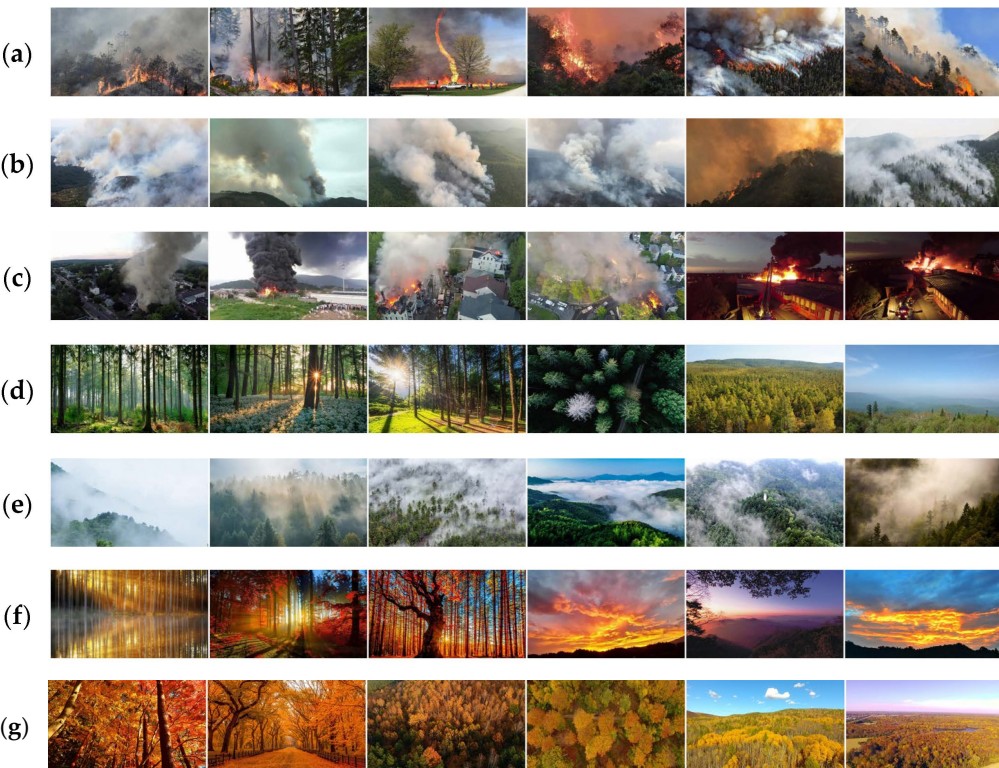

**Figure 1.** Example images from the dataset, including images of fire and smoke in forests and cities, typical forest images, and interference images. (**a**–**c**) The fire images mainly come from various scenes such as forests, wilderness areas, and urban areas, depicting flames, white smoke, black smoke, dense smoke, and other associated phenomena during fire incidents. (**d**) Non-fire undisturbed forest images were captured from various angles, including shots from inside the forest and aerial views from above the forest. (**e**–**g**) Non-fire images are accompanied by various disruptive elements, including clouds, sunlight, tree leaves with colors similar to fire, and pyrocumulus clouds.

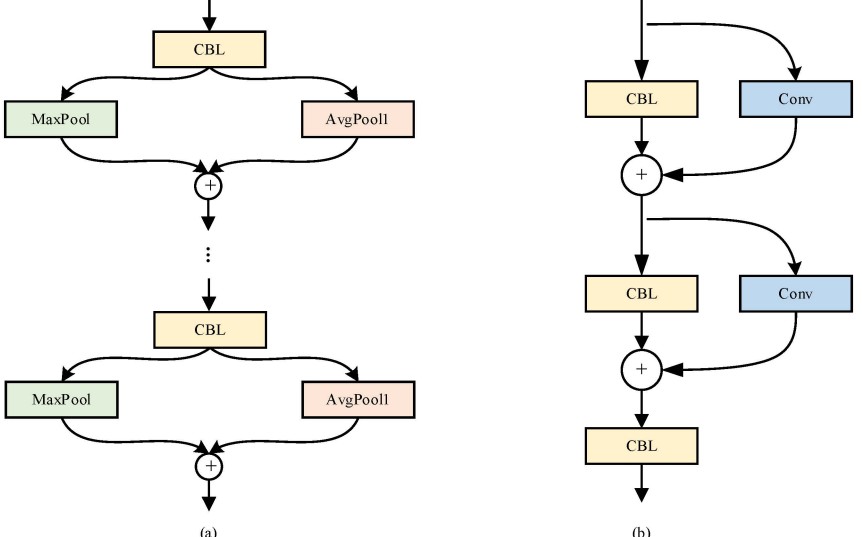

**Figure 2.** Network architectures for two feature fusion methods. (**a**) The first feature fusion network structure, inspired by the SPP structure. (**b**) The second feature fusion network structure, inspired by the residual structure.

### 2.2.2. Transfer Learning

Training a deep learning model for a specific task is a complex and expensive process that involves various challenges. Firstly, creating a dataset requires a lot of effort and resources to collect and label the data accurately. Secondly, training a network from scratch is time consuming and results in the network having a slow convergence speed. More powerful hardware is required to mitigate this issue, which can be costly.

Transfer learning was proposed to address the problem of limited annotated training data. The schematic diagram of transfer learning is shown in Figure 3. Transfer learning involves transferring the model parameters that were trained on a larger, more general dataset to a new target task network [62]. By leveraging the knowledge learned from previous tasks, the model can achieve higher accuracy and faster convergence with less task-specific annotated data [63–65].

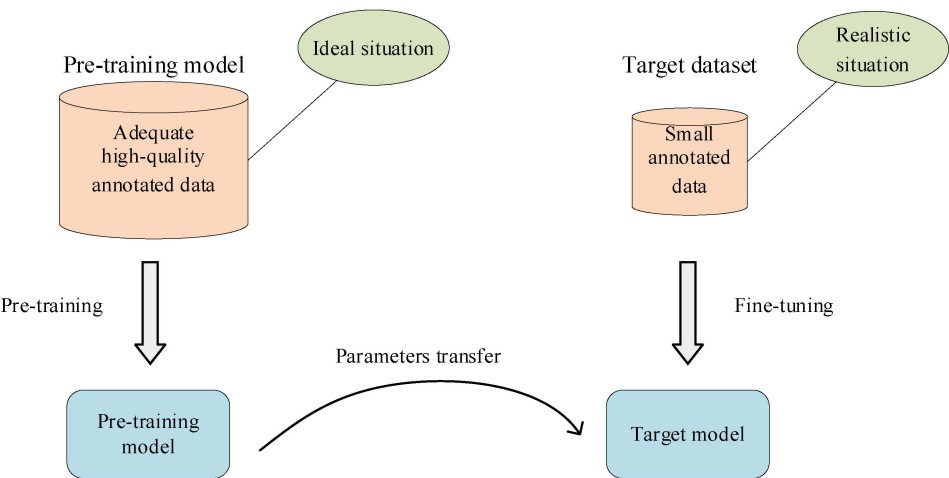

**Figure 3.** Schematic diagram of transfer learning.

The parameter migration of the single-channel network model structure in this paper is shown in Figure 4. The parameters of the C1–C2 convolutional layers and the parameters of the three fully connected layers of the pre-trained Alexnet model on the Imagenet dataset are migrated to the modified single-channel model and then trained using the fire dataset in this paper.

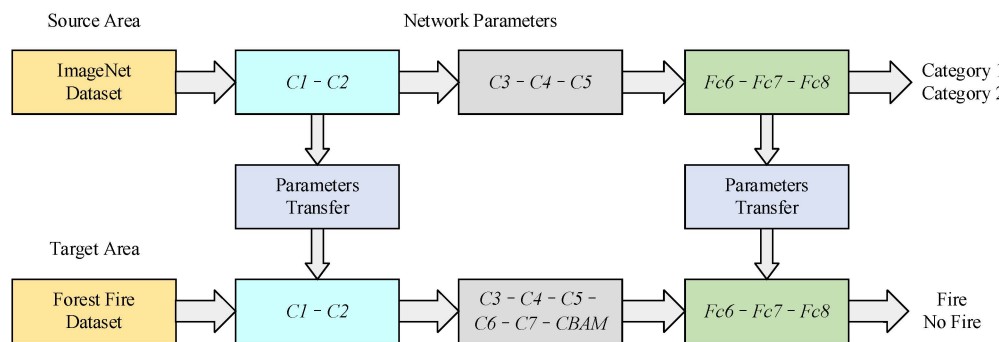

**Figure 4.** Schematic diagram of migration learning for single-channel network.

### 2.2.3. Attention Mechanism

The neural attention mechanism is a technique that enables neural networks to focus on input features and applies to data of various shapes. It helps the network to locate the relevant information from complex backgrounds and to suppress irrelevant information, thereby improving the network's performance and simplifying its structure.

Convolutional block attention module (CBAM) [66] is a lightweight attention mechanism network that can be seamlessly integrated into CNNs to filter out key information from a large amount of irrelevant background with limited resources and negligible overhead. The central idea of this module is to attend to the "what" along the channel axis and the "where" along the spatial axis, and to enhance the meaningful features along both dimensions by sequentially applying channel and spatial attention modules. By doing so, the module can capture the most relevant information in both the channel and spatial dimensions, resulting in improved feature representations [67,68].

The detailed module structure is shown in Figure 5. The parameter settings for the CBAM module are shown in Table 1.

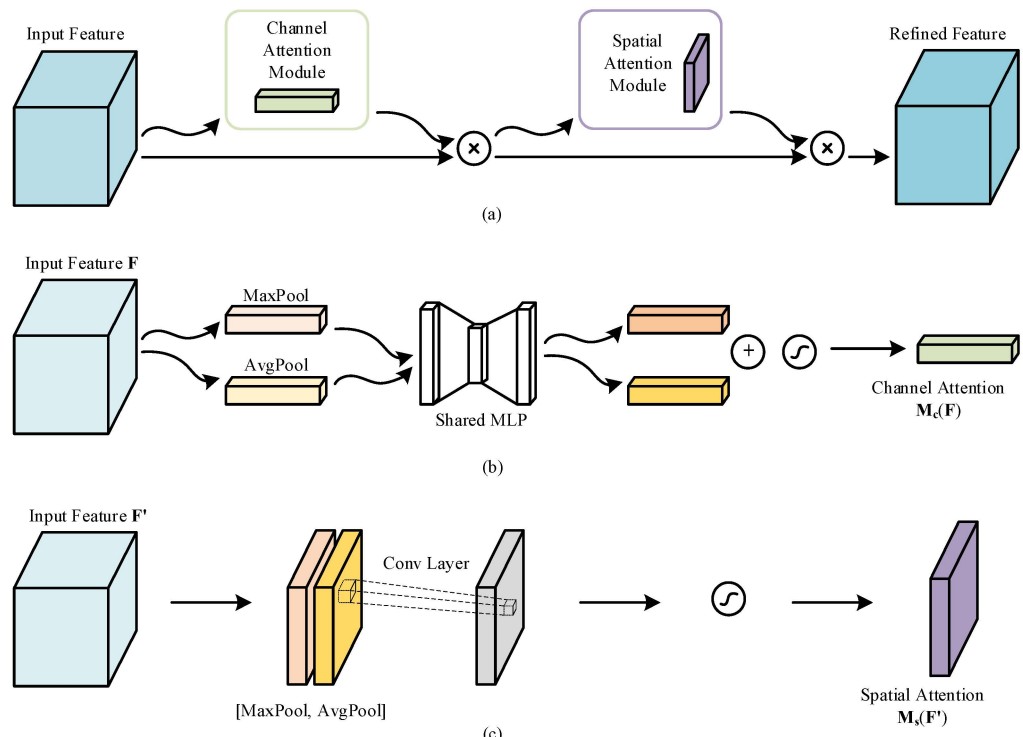

**Figure 5.** (**a**) Structure of the CBAM, (**b**) structure of the channel attention module, (**c**) structure of the spatial attention module.

**Table 1.** The parameter settings for the CBAM module.

| Module | Type | Input Size | Kernel Size | Kernel Number | Output Size | Stride | Padding |
|---|---|---|---|---|---|---|---|
| Channel | MaxPool | $H \times W \times C$ | $H \times W$ | None | $1 \times 1 \times C$ | 1 | None |
| | AvgPool | $H \times W \times C$ | $H \times W$ | None | $1 \times 1 \times C$ | 1 | None |
| | Fc1 | $C$ | None | None | $C/16$ | None | None |
| | Fc2 | $C/16$ | None | None | $C$ | None | None |
| Spatial | Conv1 | $H \times W \times 2$ | $7 \times 7$ | 1 | $H \times W \times 1$ | 1 | 3 |
| | Pool1 | $56 \times 56 \times 64$ | $3 \times 3$ | None | $27 \times 27 \times 64$ | 2 | 0 |

The calculation formulas for the channel attention module and spatial attention module are shown in Equations (1) and (2), respectively.

$$M_c(F) = \sigma(MLP(MaxPool(F)) + MLP(AvgPool(F))) \tag{1}$$

$$M_s(F') = \sigma\left(f^{7\times7}\left([AvgPool(F'); MaxPool(F')]\right)\right) \tag{2}$$

Assuming that the size of the inputs $F$ and $F'$ for both the channel attention module and spatial attention module is $R^{C \times H \times W}$, the corresponding output sizes for $M_c$ and $M_s$ are

$R^{C \times 1 \times 1}$ and $R^{1 \times H \times W}$, respectively. The notation $f^{7 \times 7}$ represents a convolution operation with a $7 \times 7$ kernel size, where $\sigma$ denotes the sigmoid function.

### 2.3. Establishment of an Improved Single-Channel Model

The classical Alexnet network [69] is chosen as the base network, which consists of 5 convolutional layers, 3 pooling layers, and 3 fully connected layers.

The improvement of the single-channel model is divided into two main aspects. On one hand, there is feature fusion. One approach is to perform global average pooling and global maximum pooling operations on the extracted features, and then combine the results to improve feature characterization. In this case, the two pooling layers within the same layer share the convolution kernel size, step size, padding, and other parameters. The second fusion uses convolutional layers to combine shallow and deep features. Conv6 and Conv7, a $1 \times 1$ convolutional layer, are added in parallel at the location of the original network Conv3 and Conv4, respectively, for the fusion of shallow and deep features. The Conv6 layer adjusts the output of the Pool2 layer to $13 \times 13 \times 384$ and then fuses it with the output of the Conv3 layer. Conv7 processes the first feature fusion result and then fuses it with the result of the Conv4 layer.

The structure diagram of the improved single-channel model is shown in Figure 6.

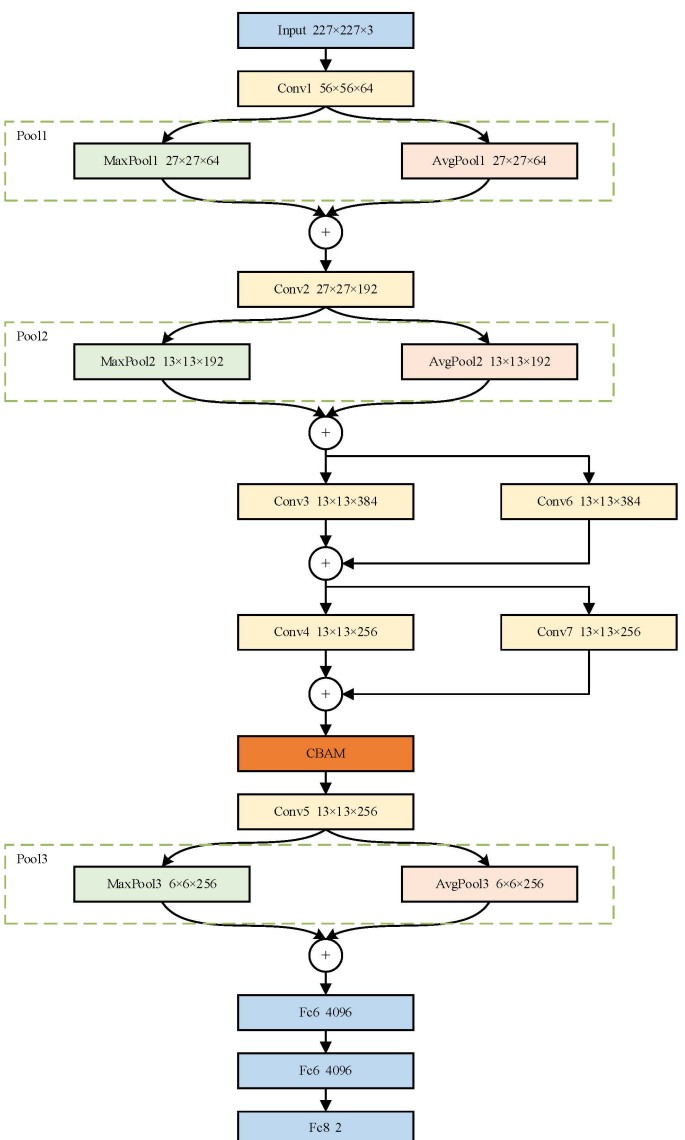

**Figure 6.** Structure of the improved single-channel network.

*2.4. Establishment of a Novel Dual-Channel Network*

In this part, a two-channel CNN model that can cover different-sized fire scenes is designed. First, another network with structurally similar inputs of 336 × 336 × 3 is designed based on the improved single-channel model. Then, the feature extraction results of the two networks are fused to obtain a novel two-channel network, whose structure is shown in Figure 7.

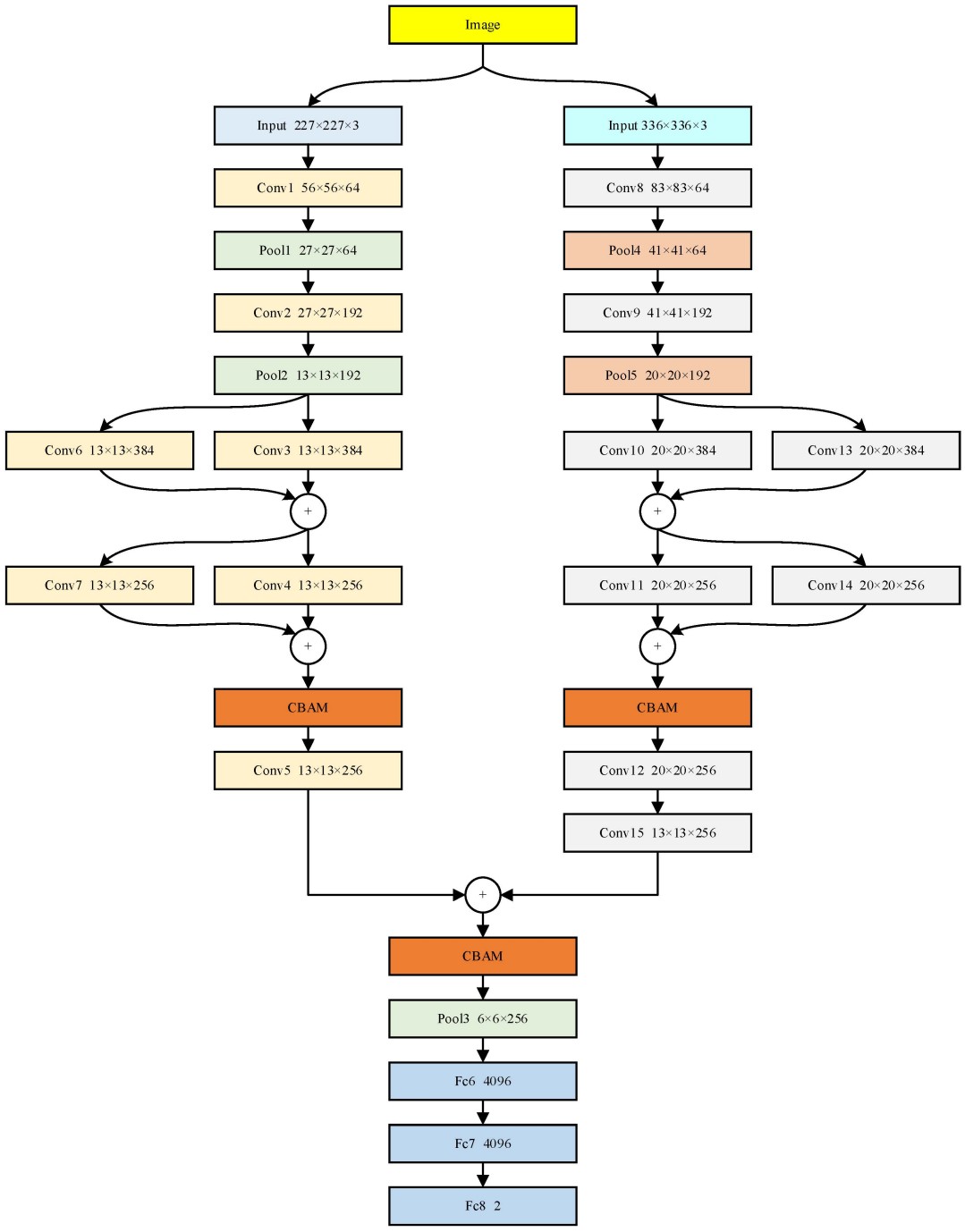

**Figure 7.** Structure of the novel dual-channel network.

To facilitate the distinction, the two single-channel networks are renamed; the channel network where the input is 227 × 227 × 3 is called the first channel, and the one with the input of 336 × 336 × 3 is called the second channel. The parameters of the novel dual-channel network are shown in Table 2.

**Table 2.** Parameters of the novel dual-channel network.

| Type | Input Size | Kernel Size | Kernel Number | Output Size | Stride | Padding |
|------|-----------|-------------|---------------|-------------|--------|---------|
| Conv1 | $227 \times 227 \times 3$ | $11 \times 11$ | 64 | $56 \times 56 \times 64$ | 4 | 2 |
| Pool1 | $56 \times 56 \times 64$ | $3 \times 3$ | none | $27 \times 27 \times 64$ | 2 | 0 |
| Conv2 | $27 \times 27 \times 64$ | $5 \times 5$ | 192 | $27 \times 27 \times 192$ | 1 | 2 |
| Pool2 | $27 \times 27 \times 192$ | $3 \times 3$ | none | $13 \times 13 \times 192$ | 2 | 0 |
| Conv3 | $13 \times 13 \times 192$ | $3 \times 3$ | 384 | $13 \times 13 \times 384$ | 1 | 1 |
| Conv4 | $13 \times 13 \times 384$ | $3 \times 3$ | 256 | $13 \times 13 \times 256$ | 1 | 1 |
| Conv5 | $13 \times 13 \times 256$ | $3 \times 3$ | 256 | $13 \times 13 \times 256$ | 1 | 1 |
| Conv6 | $13 \times 13 \times 192$ | $1 \times 1$ | 384 | $13 \times 13 \times 384$ | 1 | 0 |
| Conv7 | $13 \times 13 \times 384$ | $1 \times 1$ | 256 | $13 \times 13 \times 256$ | 1 | 0 |
| Conv8 | $336 \times 336 \times 3$ | $11 \times 11$ | 64 | $83 \times 83 \times 64$ | 4 | 2 |
| Pool4 | $83 \times 83 \times 64$ | $3 \times 3$ | none | $41 \times 41 \times 64$ | 2 | 0 |
| Conv9 | $41 \times 41 \times 64$ | $5 \times 5$ | 192 | $41 \times 41 \times 192$ | 1 | 2 |
| Pool5 | $41 \times 41 \times 192$ | $3 \times 3$ | none | $20 \times 20 \times 192$ | 2 | 0 |
| Conv10 | $20 \times 20 \times 192$ | $3 \times 3$ | 384 | $20 \times 20 \times 384$ | 1 | 1 |
| Conv11 | $20 \times 20 \times 384$ | $3 \times 3$ | 256 | $20 \times 20 \times 256$ | 1 | 1 |
| Conv12 | $20 \times 20 \times 256$ | $3 \times 3$ | 256 | $20 \times 20 \times 256$ | 1 | 1 |
| Conv13 | $20 \times 20 \times 192$ | $1 \times 1$ | 384 | $20 \times 20 \times 384$ | 1 | 0 |
| Conv14 | $20 \times 20 \times 384$ | $1 \times 1$ | 256 | $20 \times 20 \times 256$ | 1 | 0 |
| Conv15 | $20 \times 20 \times 256$ | $1 \times 1$ | 256 | $13 \times 13 \times 256$ | 2 | 3 |
| Pool3 | $13 \times 13 \times 256$ | $3 \times 3$ | none | $6 \times 6 \times 256$ | 2 | 0 |
| Fc6 | 9216 | none | none | 4096 | none | none |
| Fc7 | 4096 | none | none | 4096 | none | none |
| Fc8 | 4096 | none | none | 2 | none | none |

It should be noted that the output of the second channel at the Conv12 layer is $20 \times 20 \times 256$, which is different from the output size of the Conv12 layer of the first channel. Therefore, the two need to be resized before fusion. In this paper, we chose to build a $1 \times 1$ convolutional unit (Conv15), and the output of the Conv12 layer was adjusted to $13 \times 13 \times 256$. After the fusion of the features of the two channels, it is still necessary to process them using CBAM and then continue with operations such as pooling layers (Pool3) and fully connected layers (Fc6–Fc7–Fc8).

## 3. Results

The experiment was carried out in the Pytorch framework of Windows 10, using an Intel® Core™ i7-12700k CPU (Santa Clara, CA, USA) running at a standard frequency of 5.00 GHz, 128 GB of RAM, and a GPU with an NVIDIA RTX 3090 (Santa Clara, CA, USA) acting as the hardware gas pedal for model training.

### 3.1. Simulation Analysis of the Improved Single-Channel Model

The image size of the forest fire dataset photos is scaled to $227 \times 227 \times 3$ by scaling the transformation to meet the Alexnet network's input image size requirement. After 60 rounds of training using the resized pictures, the model's test set accuracy reaches 95.30%.

The improved single-channel model achieves superior results in the task of recognizing forest fires because the fusion of deep and shallow features enhances the feature representation. Figure 8 displays the accuracy curves of the enhanced Alexnet model before and after the enhancement of the test set. When compared to the accuracy before modification, the improved Alexnet forest fire recognition accuracy is 1.50% higher at 96.8%, and the recognition error rate drops by 31.9%.

The accuracy curves of the training set and test set of the improved single-channel model are shown in Figure 9. With a training set accuracy of 99.59% and a test set accuracy of 96.80%, the training set accuracy is higher, and the model exhibits significant overfitting.

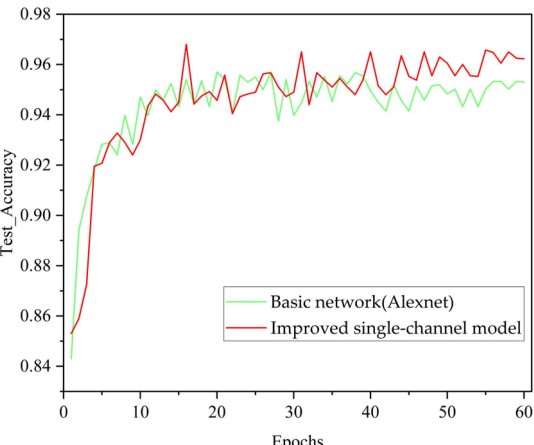

**Figure 8.** The test set accuracy curve of the basic model before and after improvement.

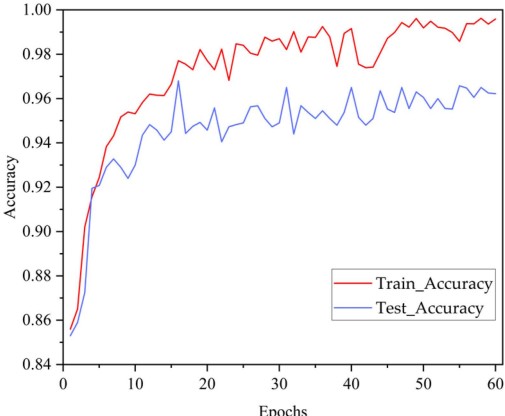

**Figure 9.** Accuracy curves of the training and test sets of the improved single-channel model.

It was demonstrated that transfer learning can address the issue of model overfitting or insufficient datasets [70–72]. This approach is also used in this study to address the ease of model overfitting and reduce the time cost. The accuracy curves of the test set of the model before and after using migration learning are shown in Figure 10, and it is clear that migration learning improved the test accuracy of the model. The accuracy of the model test set before using migration learning is 96.80%, and after using it, the accuracy is 98.45%, with a 1.65% improvement in accuracy and a 51.56% decrease in the recognition error rate.

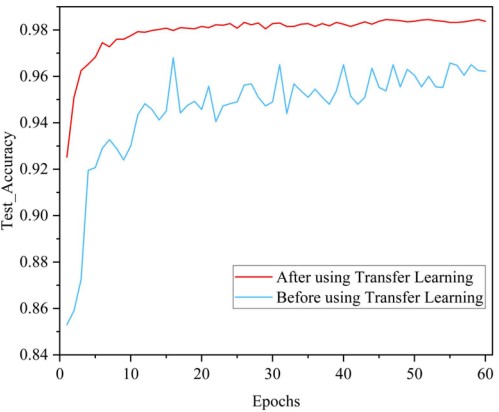

**Figure 10.** Accuracy curves of the test set before and after using migration learning.

### 3.2. Simulation Analysis of the New Dual-Channel Network

The second channel CNN with the input size of 336 × 336 × 3 is designed concerning the improved single-channel network. It is trained separately, and the accuracy of the model is 98.15% after 60 rounds of training, which is higher than the accuracy of the first channel. This indicates that designing models with different input sizes is effective in improving accuracy.

The test set accuracy of the two-channel network obtained from the fusion of the two single-channel networks is 98.90%, which is an improvement of 0.45% and 0.75% in accuracy relative to the first and second channel models, respectively. The accuracy curves of the two-channel network as well as the two separate networks on the test set are shown in Figure 11.

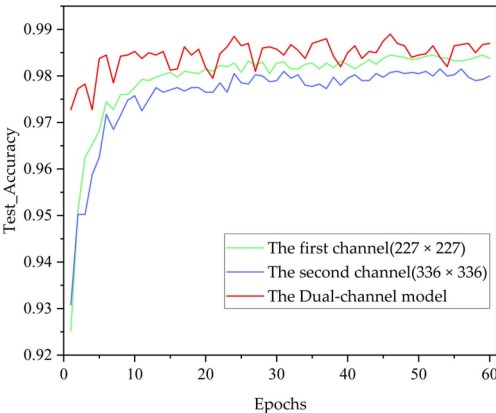

**Figure 11.** Accuracy curves of test sets for different models.

The performance of the novel two-channel model proposed in this paper is analyzed with the help of a confusion matrix. The confusion matrix is shown in Figure 12, and there are only 44 recognition errors among 4000 test samples. The model has a strong performance with an accurate prediction rate of 98.9%, a precision of 99.24%, a recall rate of 98.55%, and a specificity of 98.56%. An inspection of the misidentified images shows that the model misidentifies images of heavy clouds, images of fire clouds, and images of forests with large fire-like colors.

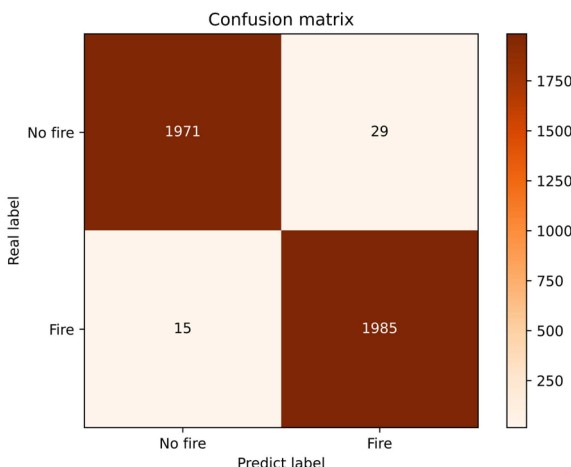

**Figure 12.** The confusion matrix of the dual-channel CNN.

Additionally, this study contrasts VGG16 [73] and Resnet50 [74], which are two different deep-learning methods. The test results are provided in Table 3, and the model in this research has the greatest accuracy of 98.90% among these methods.

**Table 3.** Performance comparison between different models.

| The Relevant Literature | Models | Model Accuracy (%) |
|---|---|---|
| This paper | Novel dual-channel CNN | 98.90 |
| [73] | VGG16 | 95.88 |
| [74] | Resnet50 | 97.78 |

## 4. Discussion

As shown in Table 4, the performance of the two-channel network model previously proposed in this paper and the subject group [47] is compared. Obviously, the new two-channel network outperforms the previous network in terms of accuracy, precision, and recall. It is worth mentioning that the recognition error rate of the new two-channel network is reduced by 28.89% compared with the previously proposed two-channel network, in which the probability of misclassifying a fire as a non-fire is reduced by 48.28%.

**Table 4.** Performance comparison between dual-channel CNNs.

| The Literature | TP | FN | FP | TN | Accuracy | Precision | Recall |
|---|---|---|---|---|---|---|---|
| This paper | 1971 | 29 | 15 | 1985 | 98.90 | 99.24 | 98.55 |
| [47] | 1968 | 32 | 29 | 1971 | 98.48 | 98.55 | 98.40 |

The model described in this research is also compared to other models in the field. The RMSN proposed in [75] is a fire detection model based on RNN architecture. It combines temporal and spatial network features to achieve enhanced accuracy in fire detection. However, the use of RNNs in this model results in slower detection speeds and increased hardware requirements.

Furthermore, the dataset itself is an important limiting factor. The dataset utilized in [75] focuses solely on fire smoke images in different scenarios, disregarding the consideration of other potential interferences such as clouds, fog, or fire-like objects and their impact on fire detection. This limitation is not only evident in [75] alone, but also in [76,77], and in the other related literature. These datasets predominantly consider the influence of specific factors while neglecting the comprehensive consideration of multiple factors. However, the dataset in this study is more comprehensive. In addition to the typical forest fire data, it also includes a wide range of interfering factors such as clouds, sunlight, fire clouds, and forest areas with fire-like colors. It is foreseeable that in future research on deep-learning-based or machine-learning-based forest fire identification methods, there will be an increased focus on improving the dataset in addition to enhancing the network structure.

## 5. Conclusions

Forest fires have the potential to negatively impact forest ecosystems, cause economic losses, and even pose a threat to human lives. Therefore, the development of an effective fire detection method is crucial. In this paper, a novel two-channel network forest fire identification method is proposed that achieves higher accuracy in fire detection. Two different feature fusion approaches are utilized to combine features at various stages of the underlying network, thereby enhancing the characterization capability of the features. Subsequently, the enhanced features are streamlined, key information is extracted, and feature redundancy is reduced through the use of an attention mechanism. This process improves the effectiveness of the extracted features. And the dual-channel model exhibits superior performance as it is capable of accommodating varying input sizes. Furthermore, transfer learning plays a pivotal role in mitigating model overfitting and minimizing training time costs. The experimental results show that this new two-channel network significantly outperforms the single-channel network in fire recognition with an accuracy of 98.90%.

However, there are still some issues in particular circumstances. The thick fog in the forest might closely match the thick smoke characteristics produced during the peak stages of a fire when the weather is exceptionally foggy. Reddish-hued forests are already prone to erroneous interpretation; sunlight interference makes this risk even worse. Given that both fog and sunlight frequently occur in the actual world, future research on image-based forest fire monitoring should pay particular attention to these two sources of interference. To further improve the model, we intend to increase the quality and quantity of wildfire photographs, with a special focus on examining fog properties, sunshine during various seasons, and leaf traits that resemble fire colors. In order to increase the model's accuracy and resilience, we will also add more difficult images that can result in identification failures to the training dataset.

**Author Contributions:** Conceptualization, Y.G. and Z.Z.; methodology, Z.Z.; software, Z.Z.; validation, G.C. and Y.G.; formal analysis, Z.Z.; investigation, G.C.; resources, Z.X.; data curation, Z.Z. and G.C.; writing—original draft preparation, Z.Z.; writing—review and editing, Y.G.; visualization, G.C.; supervision, Y.G.; project administration, Z.X.; funding acquisition, Y.G. All authors have read and agreed to the published version of the manuscript.

**Funding:** This research was funded by the National Program on the Key R&D Project of China, grant number 2020YFB2103500, and the Major Project of Fundamental Research on Frontier Leading Technology of Jiangsu Province, grant number BK20222006.

**Data Availability Statement:** Not applicable.

**Conflicts of Interest:** The authors declare no conflict of interest.

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
