# Peer review of "Wildfire Detection via a Dual-Channel CNN with Multi-Level Feature Fusion"

_forests, doi:10.3390/f14071499_

Round 1

Reviewer 1 Report

developing Deep Learning techniques. However, I have some slight qualms with the format of the document.

Section 2.2 can be entirely distinct as a "Background" section.

The applied methodology is unclear, given the connection between the basic concepts in section 2.2 and how to implement them within the work. The ideal would be to define it in another paragraph or section to support it with a graphic that allows the reader to understand more directly the general steps performed in generating and validating the network. 

Reviewer 2 Report

Dear and respective authors,

The manuscript entitled ” Wildfire Detection via a Dual-Channel CNN with Multi-Level Feature Fusionrepresents a valuable study where a novel dual-channel CNN for forest fire is proposed based on multiple feature enhancement techniques. The used approach can have solid practical implications.

Although the presented work with valuable methodology and results deserve to be considered for publishing in the Forests scientific journal, it still has some issues needed to be addressed before this step. Below is the list with my suggestions for manuscript enhancement.

1. Abstract: page 1, lines 19-20: the authors should mention practical implications of the applied research: e.g. ”The initial results from this study can be used for creation of the platform for fire wildfire management operational framework that can be used as a tool for decision making prevention, detection and monitoring of fire hazards at different levels”.

2. Keywords: please add ”wildfire hazard” as the first keyword.

3. In the Introduction part, respective authors should expand paragraphs on page 1, lines 24-35, where more information about fire hazards on different levels are needed. You can check some useful literature such as:

Rosavec, R.; Barčić, D.; Španjol, Ž.; Oršanić, M.; Dubravac, T.; Antonović, A. Flammability and Combustibility of Two Mediterranean Species in Relation to Forest Fires in Croatia. Forests 2022, 13, 1266. https://doi.org/10.3390/f13081266

Lukić, T.; Marić, P.; Hrnjak, I.; Gavrilov, M.B.; Mladjan, D.; Zorn, M.; Komac, B.; Milošević, Z.; Marković, S.B.; Sakulski, D.; et al. Forest fire analysis and classification based on Serbian case study. Acta Geogr. Slov. 2017, 57, 51–63.

Tošić, I.; Mladjan, D.; Gavrilov, M.B.; Zivanović, S.; Radaković, M.G.; Putniković, S.; Petrović, P.; Mistridzelović, I.K.; Marković, S. Potential influence of meteorological variables on forest fire risk in Serbia during the period 2000–2017. Open Geosci. 2019, 11, 414–425.

Novkovic, I.; Marković, G.B.; Lukić, D.; Dragičević, S.; Milošević, M.; Djurdjić, S.; Samardžić, I.; Lezaić, T.; Tadić, M. GIS-Based Forest Fire Susceptibility Zonation with IoT Sensor Network Support, Case Study—Nature Park Golija, Serbia. Sensors 2021, 21, 6520.

4. Page 2, lines 52-59: citations are needed for the entire written paragraph.

5. On page 3, line 103: please clearly outline the main goals of the given research with possible practical applications.

6. On page 3, chapter 2. Materials and Methods (lines 105 onwards): more information about the used dataset is needed. Did the authors have the training/test dataset composed of different classes: i.e. artificial areas, agricultural areas, grassland, shrub, broad-leaved forest, mixed forest, coniferous forest, and barren soil/rocks? This needs to be explained in more detail. How were the interference factors being handled, what approaches the authors have used? This part is very poorly written, with absolutely not a single reference or literature being called upon. Also, there is no indication on the geographical region of the selected images with its main physical and climate properties.

7. Discussion part is completely omitted. The authors should make an effort to try and compare their results with similar studies worldwide in order to clearly outline strengths and limitations of the used methodology respectively.

8. In the conclusion part, please clearly outline strengths and limitations of the used novel approach with more emphasis on the possible practical implications in future studies.

9. Reference list must be enhanced.

Kind regards!

 Author Response

Round 2

Reviewer 2 Report

Dear and respective authors,

After thorough inspection of the revised manuscript I can confirm that it was enhanced according to the reviewers comments. Therefore, the overall quality of the paper was increased. 

Best regards!